# Behavioural Nudges, Physico-Chemical Solutions, and Sensory Strategies to Reduce People’s Salt Consumption

**DOI:** 10.3390/foods11193092

**Published:** 2022-10-05

**Authors:** Charles Spence

**Affiliations:** Department of Experimental Psychology, New Radcliffe House, University of Oxford, Oxford OX2 6BW, UK; charles.spence@psy.ox.ac.uk

**Keywords:** salt-reduction strategies, gastrophysics, aroma, flavour, multisensory, congruency

## Abstract

This narrative historical review examines the wide range of approaches that has been trialled/suggested in order to reduce the consumption of salt. While sodium is an essential micronutrient, there is widespread evidence that high levels of consumption are leading to various negative health outcomes. This review summarises the evidence relating to the various approaches that have been put forward to date to help reduce salt consumption over the years, while also highlighting a number of important questions that remains for future research. Solutions to reducing salt consumption include everything from the gradual reduction in salt in foods through to the reduction in the number/size of holes in saltshakers (what one might consider a behavioural nudge). Physico-chemical solutions have included salt replacers, such as monosodium glutamate (MSG) through to the asymmetric distribution of salt in processed (e.g., layered) foods. A wide range of sensory approaches to modulating expected and perceived saltiness have also been suggested, including the use of salty aromas, as well as suggesting the use of colour cues, sonic seasoning, and even textural primes. It is currently unclear whether different salty aromas can be combined to increase odour-induced taste enhancement (OITE) effectiveness. In the years ahead, it will be interesting to assess how long such solutions remain effective, as well as whether different solutions can be combined to help reduce salt consumption without having to compromise on taste/flavour

## 1. Introduction

Salt is an essential mineral in the human diet and has long been used to help season and, thus, add flavour to food [1]. Since pre-history, salt has also played an important preserving function (e.g., in the case of salting fish and meat) [2,3]. It has been suggested that salt has occupied a place of culinary dominance across cultures for centuries [4]. According to Rachel Herz [5]: “We like the taste of salt innately because salt is a signal of protein in nature” (quoted in [6]). Others have argued, perhaps more persuasively, that the allure of salt taste is that it signals the presence of essential minerals instead [7]. Interestingly, however, research from van Dongen and colleagues has revealed that in a range of 50 commonly consumed foods, the presence of both salty and savoury tastes was independently correlated with the presence of both sodium and protein [8]. Humans need a small amount of salt to regulate fluid balance and help the nerves and muscles to function properly. The preference/need for salt is, however, not uniquely human. Famously, the monkeys at Koshima Inlet on an island off the coast of Japan salt sweet potatoes by washing them in sea water prior to consuming them [9,10]. Amazonian parrots flock to certain cliffs (mineral licks) to get the sodium and other nutrients they need, which is otherwise hard to come by in their diet [11,12,13]. It is commonly believed that providing salt licks for deer can give rise to improved antler quality [14]. Dogs, meanwhile, are fairly insensitive to the taste of salt, the argument being that meat, which would have featured heavily in their ancestral carnivorous diet, would naturally have been high in sodium [15,16].

Salt is both a very successful flavour enhancer and can also mask the taste of bitterness [17,18]. While salt is by no means the only basic tastant to perform such a function—umami, for example, has also been reported to suppress other tastes, while, at the same time enhancing the tastes of sweet and salt [19,20] and, therefore, it does a better job than any other tastant. However, while salt is undoubtedly an essential nutrient in the human diet, there is mounting evidence of the negative health consequences that are associated with the overconsumption of sodium. This, in turn, has led to growing calls for innovative salt-reduction strategies to help improve public health outcomes [21], strategies that go beyond merely taxing high-sodium foods as has been trialled in some countries [22]. 

Over the years, several different approaches have been proposed, from physico-structural solutions in the case of processed foods, through to various kinds of nudges, such as reducing the hole sizes in salt dispensers. Furthermore, everything from aromas to colours and from certain sounds to particular tactile/oral-somatosensory attributes have been shown to modulate expected, and often also perceived, saltiness, at least under carefully controlled laboratory conditions. The key question to be addressed in this review is how effective these various approaches to salt reduction might be in the long term in the real world and whether different approaches to salt reduction can be combined effectively. Finding long-term solutions to reduce salt intake is especially important given that salt warning and nutritional labels tend to have little effect [23,24]. In this narrative historical review, I summarize the evidence relating to the various approaches that have been put forward to date to help reduce salt consumption, while also highlighting a number of important questions that remain for future research. 

## 2. Negative Health Consequences of the Overconsumption of Salt

Despite the fact that small amounts of salt are essential for healthy nutrition, a growing body of robust scientific evidence has demonstrated that the overconsumption of salt in our diets is leading to a number of negative health consequences, including hypertension (i.e., high blood pressure), which increases the risk of heart disease and stroke [25], currently two of the leading causes of mortality. At the outset, though, it is important to note how the salt we consume comes not only from the foods that we choose to eat, but also from the sodium that is added at the table, either in the form of table salt (see [26,27] on the origins of table salt) or as a condiment, such as soy sauce, fish sauce, or Maggi-type seasoning [28]. Given the ready availability of table salt and salty condiments at the dining table, the danger is that simply reducing the salt levels in popular food products might well result in consumers adding salt (or salty condiments) to enhance the taste at the table [29]. Here it is intriguing to note that there have been some legislative efforts to remove salt from the dining tables of schools and kindergartens [30]. 

Bear in mind here that just a single tablespoon of mayonnaise contains around 11 g of fat, 100 cal, and 85 mg of sodium [31]. Some of the most shocking epidemiological evidence to have emerged recently suggests that those who regularly add salt to their food lower their life expectancy at 50 years of age by an average of 1.5 years for women and 2.3 years for men, relative to those who do not salt their food [25]. This was the result of a study of more than half a million participants from the UK biobank who were followed-up for a median of 9 years. While the current recommended daily intake of salt is 5–6 g [32,33], typical consumption figures are closer to double that, depending on the culture/country [34,35,36]. This contrasts markedly with the diets of our pre-agricultural ancestors, which may well have been low in salt, with estimated daily sodium intake of 768 mg, equivalent to a little under 2 g of salt [37].

While the rhetoric in the popular science press in recent years has been to blame the food industry for loading processed foods with sugar, salt, and fat [38,39], it should be noted that saltiness is not always a desirable characteristic. Indeed, food companies have been actively trying to suppress the salty taste of isotonic drinks to help make their beverages more palatable. At the same time, however, given the flavour-enhancing qualities of salt, reducing its presence in processed foods often results in an unacceptable loss in taste quality, leading to a backlash from consumers when the salt level of their favourite brands is reduced suddenly [40,41,42,43]. What is more, there are certain traditional foods, such as bread, where salt plays an essential structural function. In the latter case, the research shows that the addition of salt influences dough elasticity, improves taste, and also enhances the browning of the crust [44,45,46]. Salt is also a key ingredient in cheese making [47]. At the same time, however, there is a growing awareness of the exceptionally high levels of salt in some cheeses, such as Roquefort [48], where a 30 g serving contains more salt (1.06 g) than a small bag of crisps/potato chips. It has been estimated that more than 20–30% of dietary salt comes from meat and meat products [49] and that around 80% of salt consumed is hidden in processed, canteen, restaurant, and fast food [50]. Salt also provides an important role in helping to preserve meat products. Consequently, any attempt to reduce the salt levels may give rise to increased issues with food safety [51].

### What Determines the Preferred Salt Level?

There is some debate as to whether our salt preferences are hard-wired or learnt. For instance, one large global study (of more than 10,000 men and women over a 24 h period) found very similar levels of salt intake across a wide range of countries/cultures [52], thus, suggesting an innate preference. There is widespread evidence for an innate hunger for salt across many species, with salt hunger presumably helping to ensure a sufficient intake of sodium [53,54]. At the same time, however, other evidence shows that those on restricted sodium diets soon adapt to the reduced saltiness in their food [55]. In particular, Gary Beauchamp and colleagues conducted a study in which people put on a controlled, low-sodium diet showed taste adaptation. Specifically, within a period of one to two months, the amount of salt that the participants found optimal in soup or crackers declined by 40 or 50 percent. Put the other way, the more salt people eat, the more they crave it. The way in which people taste their food has also been shown to influence perceived saltiness [56]. Separately, there has been an ongoing debate in the food science literature about how best to measure salt thresholds [57,58]. Bear in mind here only how in perception threshold studies, participants are normally presented with water-based pure salt solutions, whereas most food products represent complex matrices, which may result in multi-factor perceptual interactions. One of the challenges here in a real-world context is that the way in which a food is labelled (or described) can also modulate rated saltiness as a function of expectations or even disconfirmed expectations [59,60].

Interestingly, infants only start to respond to the taste of salt at around six months of age [61,62]. At the other end of the age spectrum, the declining chemosensory abilities that have been extensively documented in the elderly can all too easily lead to unhealthy eating habits [63,64], as the latter increase their intake of salt and sugar to make up for their inability to taste these ingredients in food at lower concentrations [65,66]. According to Stevens and colleagues [67], older individuals may need to add as much as two- or three-times more salt to perceive the same intensity in a tomato soup as those who are younger. Worryingly, this figure was found to increase to twelve-times as much salt added for those older individuals who were on five or more medications, which turns out to be the majority of them [68]. Given the negative health consequences of the overconsumption of salt (e.g., hypertension), this likely represents a very serious issue and one that needs to be tackled by those hoping to optimise food delivery amongst the elderly. Note here that according to observational data from the Framingham study in North America, the lifetime risk of developing hypertension in those who are 55–65 years of age is 90% [69].

## 3. Behavioural Nudging to Modify Salt-Related Food Behaviours

In recent years, there has been growing interest in the potential use of a range of sensory nudges to influence consumer behaviour in the context of food and drink [70,71]. “Nudging” refers to the use of more-or-less subtle sensory contextual cues that are designed to encourage people to behave in ways that are likely to help in improving the health, social support welfare, sustainability, and/or happiness of the individual and/or the society in which they live. One strategy that has worked effectively to reduce salt levels in processed foods has involved what has been referred to as a ‘health-by-stealth’ approach [72]. Cereal companies have been gradually decreasing the amount of salt in their breakfast cereals. Overall, they have managed to reduce the salt content by up to 47% (between 1992 and 2015), without any noticeable customer backlash, simply by doing it so gradually that customers never became aware of any difference in the formulation [73]. Consequently, each successive reduction was essentially imperceptible to the consumer and, over the long term, salt levels have dropped substantially cf. [51,74,75,76,77].

It is important to note that behavioural nudges operate at different timescales, from the slow health-by-stealth approaches to reducing the salt content of, e.g., breakfast cereals, over the decades, through to the suggestion that the number and diameter of holes on saltshakers should be reduced [78,79,80]. The latter solution apparently significantly reduces the amount of salt that people add to their food, at least in the short term. Note that a similar approach has been suggested more recently in terms of reducing the use of fish sauce in university canteens serving noodles in Thailand. In particular, the results of one study indicated that a simple change in how fish sauce was served can reduce consumption. Specifically, serving fish sauce in a bowl with a spoon reduced the amount used per noodle bowl by 0.25 g, as compared to the typical situation in which fish sauce is served from a bottle [29]. Meanwhile, using a specially designed spoon with a hole induced a larger reduction of 0.58 g of fish sauce used per bowl. By contrast, the reduction in fish sauce usage associated with cognitive-/affectively oriented nudges failed to reach statistical significance. Similar solutions would presumably also help with soya sauce dispensers in Japan, given that it represents a significant source of dietary sodium for those living there [81]. According to Fukutome [82], soy sauce and other sauces containing soybean paste account for 15% of the amount of table sauce consumed in Thailand.

One of the key challenges, though, is that no other element plays such a flavour-enhancing function as salt. What is more, and as mentioned earlier, salt also plays a functional/structural role in many foods, such as bread and cheese/dairy, thus, making its effective replacement that much more challenging. Researchers have also considered the use of a range of other salt-reduction strategies. In the next section, I will review the evidence concerning various physico-chemical attempts to find salt replacers that are not derived from sodium and potential solutions associated with modifying the structural properties in processed foods.

## 4. Physico-Chemical Solutions of Salt Reduction While Maintaining Taste

There has long been interest in the possibility of finding stimuli that taste salty without relying on sodium. Molecular salt replacers, such as potassium chloride (KCl), iodized salt, etc., have been on the market for decades, often marketed as ‘low-sodium’ [83]. However, none of the other metal salts give rise to anything like as salty a taste as sodium. Furthermore, potassium chloride imparts a bitter taste if present in too high a proportion in foods [48,84]. More worryingly, an excess of potassium may cause hyperkalaemia (the name given to an abnormally high level of potassium in the blood leading to cardiac arrest), which is why many authorities do not recommend the use of potassium chloride as a salt substitute [85]. Others have been championing the replacement of sodium chloride with monosodium glutamate as an effective route to maintain taste/flavour, while reducing the amount of sodium consumed [86]. Note that MSG also gives rise to a somewhat salty taste [87]. Despite containing only one-third the amount of sodium (12.3 g/100 g) found in salt (39.3 g/100 g), it has been suggested that this makes MSG a particularly promising salt alternative in sodium-reduction strategies, though it is important to stress the independence of salty and umami as taste qualities, as stressed in ISO 3972. However, as will be mentioned later, some consumers show an aversion to ‘artificial’ flavour enhancers, as MSG is often labelled.

Separate to work on the development of salt replacers, a number of innovative physico-chemical food structural solutions to salt reduction have also emerged in recent years. Therefore, for example, it has been demonstrated that it is possible to maintain the perceived flavour of layered processed foods, such as lasagne, by asymmetrically distributing the salt through the food matrix rather than distributing it evenly [88,89], loading the salt taste in the first mouthful of other foods [90]. The evidence suggests that asymmetrically distributing the salt can give rise to enhanced (or maintained) taste perception while reducing the absolute quantity of salt. There may also be grounds for trying to target the salt to those parts of the oral cavity where taste receptors are more densely distributed [91,92].

Another promising approach to salt reduction involves the use of non-gustatory sensory stimuli in order to modulate the expected, and hopefully also the experienced, saltiness of foods. Therefore, for example, colours, aromas, textures, and even sonic stimuli have been shown to be associated with saltiness and can, under laboratory conditions at least, influence perceived saltiness.

## 5. Retronasal Enhancement of Saltiness Perception

One of the most popular/promising approaches to reducing salt, while, at the same time, maintaining taste, involves targeting the key role played by so-called salty aromas (see [93] for a recent review). The evidence from a growing body of empirical research demonstrates that those aromas that commonly co-occur with a salty taste in food (aromas that can be considered as congruent with a salty taste [94]) tend to take on the association with the co-occurring taste [95] and become more perceptually similar to the taste with which they commonly co-occur [96]. Therefore, for example, the odour of soy sauce and dried ham is classed as salty odour by those who are familiar with them. What is more, having become associated with a salty taste, these aromas, when added to foods give rise to odour-induced taste enhancement (OITE). This is where perceived saltiness is enhanced by the addition of odorants that are congruent (i.e., which commonly co-occur) with salty-tasting foods. At the same time, however, the presence of a specific taste (e.g., salty, umami, or sweet), has also been documented to enhance the associated food odour, no matter whether experienced orthonasally or retronasally [97,98,99].

Relevant to the theme of the present review, it has been widely suggested that OITE could be used as an effective strategy in order to enhance the perceived saltiness of reduced-salt food products. As highlighted in Table 1, many salty aromas that are clearly capable of giving rise to an OITE effect have been identified, there are several fundamental questions concerning OITE effects that remain to be addressed. 

Therefore, for example, one might wonder whether certain salty odorants are more effective than others at enhancing the perceived salty taste in food. It is currently also rather unclear whether different salty odorants can be combined to enhance their effectiveness. Recent findings also question whether heterogenous salt distribution can be combined effectively with OITE in realistic food substrates, such as hot flan [118]. The role of cultural differences in the OITE effects also constitutes an intriguing area for further study. To what extent does the aroma of specific local salty condiments come to take on enhanced OITE effects? Think here of Thai fish sauce in Thailand [29], soy sauce in Japan [81], or Maggi-type seasoning in a number of other countries [28]. Intriguingly, researchers have already started to probe the different representations of the qualities of soy sauces amongst consumers and chefs as a function of their sensory qualities cf. [119,120,121]. Another intriguing issue that has yet to be resolved is the relative importance of congruency to OITE effects (i.e., as distinct from the effects of perceptual similarity, see [94,96,99,122] on this subtle/intriguing distinction). Nevertheless, despite these outstanding issues, it is becoming increasingly clear that OITE represents a promising sensory approach to the maintenance of an acceptable taste/flavour profile in reduced-salt processed food products.

## 6. Visual Modulation of Saltiness Expectations and Perception

Beyond the olfactory contribution to salt perception, there has been much interest in the use of food colour to enhance perceived saltiness (e.g., in chicken stock/broth/bouillon [123,124,125]). In fact, a large body of research conducted over the last 90 years or so has demonstrated that colour cues in food and drink often exert a significant effect on taste thresholds [126], suprathreshold intensity ratings, and flavour-identification responses [127,128]. In an influential early psychophysical study, Maga found that adding colour (red, green, or yellow) to otherwise colourless salt solution had no impact over the salt taste detection threshold, despite the significant influence of adding colour on the threshold for sweet, bitter, and sour [126]. Other researchers though have reported that increasing the intensity of orange colour from a lighter to a darker colour increased saltiness expectation (based on visual observation) in the context of mayonnaise-dipping sauce [129,130]. Wongthahan and colleagues conducted a study of soy sauces (light, medium, and dark) using both regular users (consumers) and culinary chefs. Their results highlighted an association between brown colour intensity with saltiness expectation/perception [119].

One problem, though, when it comes to the use of colour to modulate salt perception is that salty foods come in all manner of colours (this is the argument put forward by Maga [126] to explain the null results on salt detection thresholds he reports) and, hence, it has been suggested that colour may be a less effective cue to saltiness perception that in the case of these colours associated with other tastes (e.g., consider the association between pinkish-red and sweetness; [131,132]; see also [133]). While early research tended to focus on the association between orangey-brown colour of savoury stocks and broth [29,123,124,134,135] and brown colour of condiments/seasonings, such as fish sauce and soy sauce [119], contemporary research on the cross-modal correspondences that exist between colour patches and taste have highlighted a robust association between the colours white and blue (either when presented individually or else together) and a salty taste [136,137,138,139]. While the link between salt and white may well relate to the typically white colour of salt crystals, the blue association may well be the result of packaging colour conventions in the marketplace. It should also be noted that the colours orange/brown were often not available to participants in the laboratory studies of colour-taste correspondences, hence, perhaps explaining why this pairing did not appear in the results that have been reported [140].

Nevertheless, taken together, the research that has been published to date demonstrates that despite the fact that multiple colours are associated with saltiness, there may be the opportunity to enhance perceived saltiness through the use of colour cues that help to set expectations of saltiness in the consumer (cf. [141]). Further, going beyond the use of colour, there may be fruitful opportunities to combine salty colours with salty aromas in order to enhance their combined effectiveness in conveying an impression of saltiness in reduced-salt food products (cf. [131,142]).

## 7. Sonic Seasoning to Modify Saltiness Perception

Background noise has been shown to suppress the perception of salt (and sweetness) in both food products and pure taste solutions [143,144,145]. At the same time, however, the presence of food consumption sounds that are typically associated with a salty taste (such as the sound of someone noisily eating potato chips) has been shown to increase perceived saltiness [146], though note the danger in triggering misophonia in some proportion of consumers on hearing such sounds [72]. Intriguingly, sonic seasoning is where music and soundscapes with particular sonic properties are found to accentuate the corresponding taste property [147,148,149,150,151]. According to research from Wang and colleagues, the sonic qualities that are most strongly associated with saltiness were a long decay time, high auditory roughness, and a regular rhythm. Meanwhile, in terms of emotional associations, saltiness was matched with negative valence, high arousal, and minor mode [151]. A few years ago, one Beijing café even went so far as to introduce sweet sonic seasoning so that they could reduce the sugar content in their drinks without having to compromise on taste [152]. In the future, it would not seem beyond the realms of possibility to use sonic seasoning to modulate perceived saltiness in food and drink.

## 8. Tactile/Haptic Modulation of Saltiness Perception

In terms of tactile/oral-somatosensory influences on saltiness perception, early research from Christensen demonstrated an impact of the viscosity in solutions on perceived saltiness and sweetness [153], though, in this case, the mechanism behind underlying the effect might be a cross-modal correspondence between viscosity and taste or perhaps a physicochemical impact of reduced OITE due to increased viscosity. Elsewhere, Van Rompay and Groothedde [154] demonstrated that perceived saltiness of potato chips could be enhanced through the use of rough surface texture design of a serving bowl (cf. [155,156,157,158,159]). Importantly, however, these saltiness-enhancement effects were only documented for medium and full-salt crisps but not for no-salt crisps.

## 9. Challenges to Salt Reduction: The Knowledge–Behaviour Gap and Self-Efficacy

Self-efficacy refers to the perceived ability of an individual to exert personal control [160]. The first of the four main constructs in the Knowledge Behaviour Gap model is knowledge, followed by acceptance, intention, and behaviour [161]. Self-efficacy has been shown to play a major role in the maintenance of health behaviours across a variety of health domains [162]. Most chronic diseases are rooted in lifestyle factors and enhancing knowledge is useful to the extent that it subsequently leads to the modification of people’s behaviour [163,164,165]. Intriguingly, dietary modification through enhancement of self-efficacy and knowledge was a principal area of focus of the FCP (Stanford Five-City Project) campaign with some success [166]. That said, a recent study among people living in Hong Kong highlighted a worrying disengagement with salt-reduction behaviour, such as rarely/never checking the sodium or salt content listed on the food label and rarely/never purchasing food labelled as containing low-salt or no-salt [167]. Meanwhile, nearly 90% of the participants in one Greek study did not know what the recommended daily salt intake was [168]. While 90% of the participants in another study conducted in the Australian state of Victoria were aware that excessive salt intake can cause health damage, more than 80% still acknowledged that they were eating “far too much” (i.e., more than the recommended daily intake), with less than half of the participants actively attempting to reduce their salt intake [169]. Importantly, for fundamental knowledge regarding the recommended daily intake, the primary food sources of high salt content and the differences between salt and sodium continues to be lacking, even amongst those living in high-income countries [170]. According to the results of a Chinese study, promoting better public knowledge, enhancing the public’s awareness of salt reduction, and encouraging more active salt-reduction behaviour can help to suppress the transition from normal blood pressure to hypertension [171]. Ultimately, researchers believe that the population’s knowledge, attitudes, and behaviours will affect their salt consumption and are thought to be adjustable and controllable intermediate factors over the short term [172].

## 10. Conclusions

The negative health consequences of the overconsumption of salt are becoming increasingly apparent [173,174,175]. This is despite the discrepancy between the WHO recommendations that people should keep their salt intake below 5 g per day (2000 mg/day of sodium) and their potassium intake above 3500 mg/day, and the dietary reference intakes from the National Academy of Sciences, Engineering, and Medicine in the USA that the adequate intake in adults is 3.75 g/day of salt (1500 mg/day of sodium), while the adequate potassium intake is 3400 mg/day in men and 2600 mg/day in women [176]. The dangers have, for example, been highlighted by the latest epidemiological research, showing that people who salt their food have a 1.5–2.3-year reduction in average life expectancy when compared to those who do not [25]. At the same time, however, the potential health benefits of reducing salt intake (e.g., in terms of lowering blood pressure) are also becoming increasingly well-established [177,178]. In fact, effective salt-reduction strategies have been rolled out in several countries in recent years [179] and these have provided learnings for those wanting to introduce effective public health strategies around salt reduction [180]. That said, while the major sources of dietary salt have now been identified [49,50,51,181,182], one of the challenges is that unless salt reduction occurs very gradually, it can lead to a negative consumer response in terms of impaired taste/flavour perception [41,183,184,185,186]. Numerous different solutions have been proposed, including behavioural nudges, such as reducing the number/size of holes in saltshakers [78,79,80]. In recent years, researchers have increasingly been attempting to take such suggested solutions from the laboratory to increasingly realistic contexts [187,188,189]. In one recent series of four controlled laboratory studies, five-holed saltshakers were shown to deliver around 34% of the salt of 17-holed saltshakers [186]. The question of which solutions are likely to have the greatest long-lasting effect, and separately whether the various factors can be combined, is, though, an important current area for research ([118,190]).

It is currently unclear the extent to which sensory modulations of perceived saltiness, such as via OITE effects, or the use of colour cues or sonic seasoning are capable of influencing neural activity in primary taste areas [191]. Relevant here, verbal labels/descriptions have been documented to modulate primary taste areas [192,193]. That said, a colour’s effect on consumer responses can sometimes trigger a response bias [128], while even visually presented shapes can at least under certain conditions influence taste thresholds, arguing that at least some of the cross-modal effects may have genuine perceptual consequences [194]. It would seem likely that such a cross-modal modulation of salt perception is likely to be more acceptable in the long term than the electric cutlery that has been invented by Japanese researchers [195]. Indeed, it is worth noting that the various salt-reduction strategies that have been studied over the years clearly differ markedly in their ease/practicality of implementation.

According to Liem and colleagues [185], a sodium reduction of up to 30% may be acceptable in processed foods if introduced gradually (i.e., over a period of 3 years) and that the reduction can be up to 50% as long as it is in parallel with the addition of a flavour-boosting ingredient, such as soy sauce or dried bonito. The addition of chilli can also be used to help spice-up foods, as can black pepper, etc., without the negative health consequences associated with the overconsumption of salt [196]. Others have proposed that reductions in salt (and sodium) can be mitigated with monosodium glutamate (MSG) in what they have termed the ‘salt flip’ [83,197,198]. MSG is the sodium salt of L-glutamic acid. It is the most abundant amino acid in nature, constituting up to 8% to 10% of most dietary proteins, either as free glutamate or bound to other amino acids. At the same time, however, consumer concerns with the consumption of MSG (often labelled as an ‘artificial’ taste or flavour enhancer), although misplaced, continue to limit the applicability of such an approach [199,200,201,202,203,204,205,206]. The lack of familiarity of many Western consumers with umami may not help either [207,208]. Finally, here, one might consider Nico Ladenis, the Greek-born restaurateur who was known for kicking customers out of his London restaurant should they be so foolish as to ask for the salt [209] (p. 215). Given the appeal of salt, which can all too easily turn to salt hunger [210], one might wonder whether such extreme responses will ultimately be needed in order to deal with the very serious problem of the overconsumption of salt, though given that it has been estimated that more than 70% of our salt consumption comes from food consumption outside the home (i.e., from ready meals and eating out) [211,212], it is clear that, ultimately, various other strategies are also going to be needed (see also [213]).

## Figures and Tables

**Table 1 foods-11-03092-t001:** Chronological summary of published studies that have specifically studied the salt/umami taste-enhancing properties of volatile aromas. Adapted and updated from [93].

Study	Participants	Volatile Aroma	Taste Quality	Result	Comments
Murphy & Cain [100]	20 experienced p’s	Citral (citrus)	Sweet	Sig.	Aroma increased both congruent and incongruent taste intensity
			Salt	Sig.	
Frank & Byram [101]	E1: 20; E2: 20; E3: 18 untrained p’s	E1: Strawberry; E2: Peanut butter; E3: Strawberry	E1, E2: Sweet E3: Salt	Sig. n.s.	Only congruent strawberry aroma led to OITE in whipped cream base
Djordjevic et al. [102]	E1 (40 untrained p’s) E2 (same 40 p’s)	Strawberry & soy sauce	Sweet Salt	E1: Sig. E2: Sig.	Both actual and imagined odours enhanced congruent taste quality
Lawrence et al. [103]	59 untrained p’s	Range of salty food aromas including bacon and sardine	Salt	Sig.	7 salty aromas gave rise to significant enhancement in salt perception
Batenburg & van der Velden [104]	10 p’s & 2 trained panels of 10–12; Consumer panels	Chicken flavouring (inc. individual salty volatiles; e.g., sotolon)	Salt	Sig.	Untrained panel exhibited greater odour-induced salt enhancement in bouillons
Lawrence et al. [105]	27 untrained consumers	Comté cheese, sardine, & carrot	Salt	Sig.	Comté cheese & sardine aroma increased saltiness of model cheese
Nasri et al. [106]	64 untrained p’s	Sardine	Salt	Sig.	Sig. effect at low/medium salt levels, but no effect at high salt intensity
Nasri et al. [107]	61 untrained p’s	Sardine	Salt	Sig.	No effect of odour intensity on OITE
Seo et al. [108]	E1 (25 p’s); E2 (25 p’s)	Salty bacon & sweet strawberry	Salt Sweet	Sig. Sig.	Psychophysical & neuroimaging study
Manabe et al. [109]	70 p’s	Dried bonito stock	Umami	Sig.	Increased palatability of saltiness
Niimi et al. [110]	10 trained panellists	Cheese aroma mixture containing 10 volatiles	Umami Bitter	Sig. Sig.	Increasing OITE with increasing aroma intensity
Chokumnoyporn et al. [111]	10 panellists	Soy sauce	Salt	Sig.	Sig. effect on salty solutions above & below salt threshold
Emorine et al. [112]	Consumer panel (82)	Ham	Salt	Sig.	Sig. effect on salt perception
Syarifuddin et al. [113]	31 panellists	Sardine or butter	Salt/Fat	Sig.	Congruent aroma enhanced salt or fat
Kakutani et al. [114]	12 p’s	Soy sauce	Salt	Sig.	Retronasal odour after drinking, but not orthonasal odour before drinking, significantly increased salty taste
Onuma et al. [115]	E1 (12 p’s); E2 (20 p’s) E3 (12 p’s)	Soy sauce	Salt	Sig.	Saltiness enhancement demonstrated psychophysically & using neuroimaging
Manabe et al. [116]	E1 (75 p’s) E2 (75 p’s)	Soy sauce & 3-methyl-1-butanol	Salt	Sig.	Saltiness enhancement demonstrated & role of key compound identified
Sinding et al. [117]	13 untrained p’s	Beef stock	Salt	Sig.	Saltiness of reduced salt green pea soup

p’s—participants; Sig.—A significant enhancement on rated salty taste of the volatile was reported; n.s.—non-significant; OITE—odour-induced taste enhancement; inc.—including.

## Data Availability

No data is contained within the article.

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
