# Peer review of "Behavioural Nudges, Physico-Chemical Solutions, and Sensory Strategies to Reduce People’s Salt Consumption"

_foods, 2022, doi:10.3390/foods11193092_

Round 1
Reviewer 1 Report
Peer review on “Behavioural nudges, physico-chemical solutions, and sensory strategies to reduce people’s salt consumption” by Charles Spence in Foods, 2022.
Dietary salt reduction remains a dilemma in many countries. This review paper explains the behavioral nudges, physico-chemical solutions, and sensory strategies in order to reduce people’s salt consumption. Despite the fact that small amounts of salt are essential for healthy nutrition, a growing body of robust scientific evidence has demonstrated that the overconsumption of salt in our diets leads to a number of negative health consequences, including hypertension (i.e., high blood pressure), which increases the risk of heart disease and stroke. These two are currently of the leading causes of mortality. While the current recommended daily intake of salt is 5-6 g, the typical consumption figures are closer to double of it, depending on the culture and the country.
The negative health consequences of the overconsumption of salt are becoming increasing apparent. High blood pressure is one of the main causes of cardiovascular disease (CVD),
by reducing salt consumption, it lowers blood pressure and therefore decreases CVD which is the main cause of morbidity and mortality worldwide (He et al., 2020). The main contribution of this review paper the summary of the organoleptic evidence relating to the various approaches to help reduce salt consumption over the years up to date, while also highlighting a number of important questions that remain for future research.
Major comments
There are no major comments to this paper.
Minor comments
Limitations seems to be necessary to add to this review. It is not in the scope of this paper, but it is very important to mention knowledge-behavior gap and self-efficacy. Self-efficacy is the perceived ability to exert personal control (Bandura,1977). The first of the four main constructs of Knowledge Behavior Gap model is knowledge, followed by acceptance, intention, and behavior (Stibe et al, 2022). Self-efficacy also plays a major role in the maintenance of health behaviors, as has been documented across a variety of health domains (Maibach - Murphy, 1995). Most chronic diseases are rooted in lifestyle factor and enhancing knowledge is useful to the extent that it subsequently leads to behavioral modification (National Research Council, 1989; Tinsley, 1992; Rimal, 2000). Dietary modification through enhancement of self-efficacy and knowledge was a principal area of focus of the FCP (Stanford Five-City Project) campaign with some success (Farquhar et al., 1990).
It is mentioned in this paper (2.1. What determines the preferred salt level?) that a study suggests that salt consumption is an innate preference. However, other evidence shows that those on restricted sodium diets soon adapt to the reduced saltiness in their food. Disengagement with salt reduction behavior, such as rarely/never checking the sodium or salt content listed on the food label and rarely/never purchasing food labelled with low salt or no salt content, was reported in a recent study among people in Hong-Kong (Cheung et al., 2021). In Greece, nearly 90% of participants did not know the exact amount of the recommended daily salt intake (Marakis et al., 2014). In the state of Victoria/Australia, while 90% of participants were aware that excessive salt intake can cause health damage, over 80% reported that they were eating “far too much” than the recommended daily intake, with less than half of the participants attempting to reduce their salt intake (Grimes et al., 2017). Fundamental knowledge regarding the recommended daily intake, the primary food sources of high salt content, and the differences between salt and sodium continues to be lacking even in high income countries (Bhana et al., 2018). Mastering more knowledge, enhancing salt-reduction awareness and more active salt-reduction behavior can help to suppress the transition from normal blood pressure to hypertension according to a study in China (Du et al., 2022). Population’s knowledge, attitudes and behaviors are believed to affect salt consumption and are considered to be adjustable and controllable intermediate factors in a short time (Zandstra et al., 2016).
Knowledge-behavior gap and self-efficacy needs to be mentioned.
Also, it seems to be worth to mentioned, that the WHO recommends a salt intake below 5g per day (2000 mg/day of sodium) and a potassium intake above 3500mg/day. But, the dietary reference intakes from the National Academy of Sciences, Engineering, and Medicine in the USA contains that the adequate intake in adults is 3.75 g/day of salt (1500 mg/day of sodium) and adequate potassium intake is 3400 mg/day in men and 2600 mg/day in women (Oria et al., 2019).
Literature
Bandura, A. (1977): Social learning theory. Prentice Hall, Englewood Cliffs, NJ, USA
Farquhar, J. W., Fortmann, S. P., Flora, J. A., Taylor, B., Haskell, W. L., Williams, P. T., Maccoby, N., Wood, P. D. (1990): Effects of community-wide education on cardiovascular disease risk factors: The Stanford Five-City Project. Journal of the American Medical Association, 262, 359–365.
Bhana, N., Utter, J., Eyles, H. (2018): Knowledge, attitudes and behaviours related to dietary salt intake in high-income countries: A systematic review. Current Nutrition Reports, 7, 183–197.
Cheung, J., Neyle, D., Chow, P. P. K. (2021): Current Knowledge and Behavior towards Salt Reduction among Hong Kong Citizens: A Cross–Sectional Survey. International Journal of Environmental Research and Public Health, 18, 9572.
Du, X., Fang, L., Xu, J. Chen, X., Bai. Y., Wu. J., Wu. L., Zhong, J. (2022): The association of knowledge, attitudes and behaviors related to salt with 24-h urinary sodium, potassium excretion and hypertensive status. Scientific Reports, 12, 13901.
Grimes, C.A., Kelley, S.J., Stanley, S., Bolam, B., Webster, J., Khokhar, D., Nowson, C. A. (2017): Knowledge, attitudes and behaviors related to dietary salt among adults in the state of Victoria, Australia 2015. BMC Public Health, 17, 532.
He, F.J., Tan, M., Ma, Y., MacGregor, G. A. (2020): Salt Reduction to Prevent Hypertension and Cardiovascular Disease: JACC State-of-the-Art Review. Journal of the American College of Cardiology, 75, 632-647.
Maibach, E., Murphy, D. A. (1995): Self-efficacy in health promotion research and practice: Conceptualization and measurement. Health Education Research, 10, 37–50.
Marakis, G., Tsigarida, E., Mila, S., Panagiotakos, D. B. (2014): Knowledge, attitudes and behavior of Greek adults towards salt consumption: A Hellenic food authority project. Public Health Nutrition, 17, 1877–1893.
Oria, M., Harrison, M., Stallings. V. A. eds. (2019): Dietary Reference Intakes for sodium and potassium. National Academy of Sciences, Engineering, and Medicine and The National Academies Press, Washington, DC, USA
National Research Council. (1989). Committee on diet and health. Diet and health: Implications for reducing chronic disease risk. National Academy Press, Washington, DC, USA
Rimal, (2000): Closing the Knowledge–Behavior Gap in Health Promotion: The Mediating Role of Self-Efficacy. Health Communication, 12, 3, 219–237.
Stibe, A., Krüger, N., Behne, A. (2022): Knowledge Behavior Gap Model: An Application for Technology Acceptance. In: Awan, I., Younas, M., Poniszewska-MaraÅ„da, A. (eds) Mobile Web and Intelligent Information Systems. MobiWIS 2022. Lecture Notes in Computer Science, vol 13475. Springer, Cham, Switzerland
Tinsley, B. J. (1992): Multiple influences on the acquisition and socialization of children’s health attitudes and behavior: An integrative review. Child Development, 63, 1043–1069.
Zandstra, E. H., Lion, R., Newson, R. S. (2016): Salt reduction: Moving from consumer awareness to action. Food Quality and Preference, 48, 376–381.
Author Response
Reviewer 1: Peer review on “Behavioural nudges, physico-chemical solutions, and sensory strategies to reduce people’s salt consumption” by Charles Spence in Foods, 2022.
Dietary salt reduction remains a dilemma in many countries. This review paper explains the behavioral nudges, physico-chemical solutions, and sensory strategies in order to reduce people’s salt consumption. Despite the fact that small amounts of salt are essential for healthy nutrition, a growing body of robust scientific evidence has demonstrated that the overconsumption of salt in our diets leads to a number of negative health consequences, including hypertension (i.e., high blood pressure), which increases the risk of heart disease and stroke. These two are currently of the leading causes of mortality. While the current recommended daily intake of salt is 5-6 g, the typical consumption figures are closer to double of it, depending on the culture and the country.
The negative health consequences of the overconsumption of salt are becoming increasing apparent. High blood pressure is one of the main causes of cardiovascular disease (CVD),
by reducing salt consumption, it lowers blood pressure and therefore decreases CVD which is the main cause of morbidity and mortality worldwide (He et al., 2020). The main contribution of this review paper the summary of the organoleptic evidence relating to the various approaches to help reduce salt consumption over the years up to date, while also highlighting a number of important questions that remain for future research.
Thanks for positive summary.
Major comments: There are no major comments to this paper.
OK, thanks.
Minor comments: Limitations seems to be necessary to add to this review. It is not in the scope of this paper, but it is very important to mention knowledge-behavior gap and self-efficacy. Self-efficacy is the perceived ability to exert personal control (Bandura, 1977). The first of the four main constructs of Knowledge Behavior Gap model is knowledge, followed by acceptance, intention, and behavior (Stibe et al, 2022). Self-efficacy also plays a major role in the maintenance of health behaviors, as has been documented across a variety of health domains (Maibach - Murphy, 1995). Most chronic diseases are rooted in lifestyle factor and enhancing knowledge is useful to the extent that it subsequently leads to behavioral modification (National Research Council, 1989; Tinsley, 1992; Rimal, 2000). Dietary modification through enhancement of self-efficacy and knowledge was a principal area of focus of the FCP (Stanford Five-City Project) campaign with some success (Farquhar et al., 1990).
It is mentioned in this paper (2.1. What determines the preferred salt level?) that a study suggests that salt consumption is an innate preference. However, other evidence shows that those on restricted sodium diets soon adapt to the reduced saltiness in their food. Disengagement with salt reduction behavior, such as rarely/never checking the sodium or salt content listed on the food label and rarely/never purchasing food labelled with low salt or no salt content, was reported in a recent study among people in Hong-Kong (Cheung et al., 2021). In Greece, nearly 90% of participants did not know the exact amount of the recommended daily salt intake (Marakis et al., 2014). In the state of Victoria/Australia, while 90% of participants were aware that excessive salt intake can cause health damage, over 80% reported that they were eating “far too much” than the recommended daily intake, with less than half of the participants attempting to reduce their salt intake (Grimes et al., 2017). Fundamental knowledge regarding the recommended daily intake, the primary food sources of high salt content, and the differences between salt and sodium continues to be lacking even in high income countries (Bhana et al., 2018). Mastering more knowledge, enhancing salt-reduction awareness and more active salt-reduction behavior can help to suppress the transition from normal blood pressure to hypertension according to a study in China (Du et al., 2022). Population’s knowledge, attitudes and behaviors are believed to affect salt consumption and are considered to be adjustable and controllable intermediate factors in a short time (Zandstra et al., 2016).
Knowledge-behavior gap and self-efficacy needs to be mentioned.
Also, it seems to be worth to mentioned, that the WHO recommends a salt intake below 5g per day (2000 mg/day of sodium) and a potassium intake above 3500mg/day. But, the dietary reference intakes from the National Academy of Sciences, Engineering, and Medicine in the USA contains that the adequate intake in adults is 3.75 g/day of salt (1500 mg/day of sodium) and adequate potassium intake is 3400 mg/day in men and 2600 mg/day in women (Oria et al., 2019).
Great. Have added new section (Section 9) on the knowledge-behaviour gap near the end of manuscript as suggested, incorporating helpful references suggested.
Literature
Bandura, A. (1977): Social learning theory. Prentice Hall, Englewood Cliffs, NJ, USA
Farquhar, J. W., Fortmann, S. P., Flora, J. A., Taylor, B., Haskell, W. L., Williams, P. T., Maccoby, N., Wood, P. D. (1990): Effects of community-wide education on cardiovascular disease risk factors: The Stanford Five-City Project. Journal of the American Medical Association, 262, 359–365.
Bhana, N., Utter, J., Eyles, H. (2018): Knowledge, attitudes and behaviours related to dietary salt intake in high-income countries: A systematic review. Current Nutrition Reports, 7, 183–197.
Cheung, J., Neyle, D., Chow, P. P. K. (2021): Current Knowledge and Behavior towards Salt Reduction among Hong Kong Citizens: A Cross–Sectional Survey. International Journal of Environmental Research and Public Health, 18, 9572.
Du, X., Fang, L., Xu, J. Chen, X., Bai. Y., Wu. J., Wu. L., Zhong, J. (2022): The association of knowledge, attitudes and behaviors related to salt with 24-h urinary sodium, potassium excretion and hypertensive status. Scientific Reports, 12, 13901.
Grimes, C.A., Kelley, S.J., Stanley, S., Bolam, B., Webster, J., Khokhar, D., Nowson, C. A. (2017): Knowledge, attitudes and behaviors related to dietary salt among adults in the state of Victoria, Australia 2015. BMC Public Health, 17, 532.
He, F.J., Tan, M., Ma, Y., MacGregor, G. A. (2020): Salt Reduction to Prevent Hypertension and Cardiovascular Disease: JACC State-of-the-Art Review. Journal of the American College of Cardiology, 75, 632-647.
Maibach, E., Murphy, D. A. (1995): Self-efficacy in health promotion research and practice: Conceptualization and measurement. Health Education Research, 10, 37–50.
Marakis, G., Tsigarida, E., Mila, S., Panagiotakos, D. B. (2014): Knowledge, attitudes and behavior of Greek adults towards salt consumption: A Hellenic food authority project. Public Health Nutrition, 17, 1877–1893.
Oria, M., Harrison, M., Stallings. V. A. eds. (2019): Dietary Reference Intakes for sodium and potassium. National Academy of Sciences, Engineering, and Medicine and The National Academies Press, Washington, DC, USA
National Research Council. (1989). Committee on diet and health. Diet and health: Implications for reducing chronic disease risk. National Academy Press, Washington, DC, USA
Rimal, (2000): Closing the Knowledge–Behavior Gap in Health Promotion: The Mediating Role of Self-Efficacy. Health Communication, 12, 3, 219–237.
Stibe, A., Krüger, N., Behne, A. (2022): Knowledge Behavior Gap Model: An Application for Technology Acceptance. In: Awan, I., Younas, M., Poniszewska-MaraÅ„da, A. (eds) Mobile Web and Intelligent Information Systems. MobiWIS 2022. Lecture Notes in Computer Science, vol 13475. Springer, Cham, Switzerland
Tinsley, B. J. (1992): Multiple influences on the acquisition and socialization of children’s health attitudes and behavior: An integrative review. Child Development, 63, 1043–1069.
Zandstra, E. H., Lion, R., Newson, R. S. (2016): Salt reduction: Moving from consumer awareness to action. Food Quality and Preference, 48, 376–381.
All of the suggested references integrated in revision.
Reviewer 2 Report
The Author have chosen a relevant topic. Salt consumption is a public health concern; it involves behaviour related patterns, consumer expectations and hedonic value. In the introduction, there are several references to animals, which consume salt regularly. I would add the information, that in case of deers (and other related species) animals are given blocks of salt to improve their antler’s quality.
In the introduction part, it could be included, that in several countries there is a limit of salt content. If a product reaches that limit, there is an extra tax, the producer have to pay.
In Chapter 2 at the sentence citing reference 29, it might be worthy to note that there are some legislative efforts, to avoid the presence of salt on the table. That usually applies to kindergartens and schools.
At the end of Chapter 2 (after the citation of 51), I would add information on the fact that salt also have a technological effect, it is a way of preserving foods, especially meat products. If a meat product have to be produced with lower salt content, special awareness should be made to food safety issues. There are many documents available on that field; these are only examples (https://doi.org/10.1016/j.tifs.2016.10.016 and https://doi.org/10.1111/j.1541-4337.2009.00096.x)
At the end of chapter 3 we read “The challenge, though, is that no other element plays such a flavour enhancing function as salt.” I would recommend adding a few sentences here about the possible application of flavour enchancers, like monosodium glutamate (MSG). It would be also useful to clarify somewhere in the text that salty taste and umami taste are two independent taste qualities (e.g. ISO 3972). And a further concern: some consumers show aversions toward ‘artificial’ flavour enhancers – as MSG is often labelled.
„Separately, there has been an ongoing debate in the food science literature about how best to measure salt thresholds” – I strongly agree with that, since the perception threshold studies usually apply water-based pure salt solutions, while food products are complex matrices, which might cause multi-factor interactions.
Chapter 4, “Molecular salt replacers, such as potassium chloride (KCl), iodized salt, etc. have been on the market for decades, often marketed as ‘low-sodium’” – As far as I am concerned, the utilization of iodized salt is developed for such geographical areas, where the natural iodine input cannot be performed. Thus, through the consumption of iodized salt, this deficiency can be managed. Of course, with the addition of iodine, the total sodium content is reduced a bit.
Generally the paper provides a richness of information on this very important field of study.
Author Response
Reviewer 2: The Author have chosen a relevant topic. Salt consumption is a public health concern; it involves behaviour related patterns, consumer expectations and hedonic value. In the introduction, there are several references to animals, which consume salt regularly. I would add the information, that in case of deers (and other related species) animals are given blocks of salt to improve their antler’s quality.
Thanks for positive comments. Point about deers added.
In the introduction part, it could be included, that in several countries there is a limit of salt content. If a product reaches that limit, there is an extra tax, the producer have to pay.
Now mentioned additional salt tax
In Chapter 2 at the sentence citing reference 29, it might be worthy to note that there are some legislative efforts, to avoid the presence of salt on the table. That usually applies to kindergartens and schools.
Added.
At the end of Chapter 2 (after the citation of 51), I would add information on the fact that salt also have a technological effect, it is a way of preserving foods, especially meat products. If a meat product have to be produced with lower salt content, special awareness should be made to food safety issues. There are many documents available on that field; these are only examples (https://doi.org/10.1016/j.tifs.2016.10.016 and https://doi.org/10.1111/j.1541-4337.2009.00096.x)
Added mention of this important point, along with suggested references.
At the end of chapter 3 we read “The challenge, though, is that no other element plays such a flavour enhancing function as salt.” I would recommend adding a few sentences here about the possible application of flavour enchancers, like monosodium glutamate (MSG). It would be also useful to clarify somewhere in the text that salty taste and umami taste are two independent taste qualities (e.g. ISO 3972). And a further concern: some consumers show aversions toward ‘artificial’ flavour enhancers – as MSG is often labelled.
Clarified.
„Separately, there has been an ongoing debate in the food science literature about how best to measure salt thresholds” – I strongly agree with that, since the perception threshold studies usually apply water-based pure salt solutions, while food products are complex matrices, which might cause multi-factor interactions.
Agreed, have nuanced the discussion.
Chapter 4, “Molecular salt replacers, such as potassium chloride (KCl), iodized salt, etc. have been on the market for decades, often marketed as ‘low-sodium’” – As far as I am concerned, the utilization of iodized salt is developed for such geographical areas, where the natural iodine input cannot be performed. Thus, through the consumption of iodized salt, this deficiency can be managed. Of course, with the addition of iodine, the total sodium content is reduced a bit.
Mentioned.
Generally the paper provides a richness of information on this very important field of study.
Thanks for the positive comments.
Reviewer 3 Report
The manuscript deals with the solutions available to reduce salt consumption in various food products. The manuscript is well structured and clearly present the different solutions. A high number of publications is cited.
I only have few comments :
- at the beginning of paragraph 3, maybe add a sentence to remind to the reader what are nudges
- in this actual form, the table is difficult to read, maybe it should be better with a higher space between the different publications
- in the conclusion, maybe a comment about the use of these different solutions by consumers or industrials may be interesting as some are easy to use while others are more complicated
- in paragraph 2, I think "is" is missing in the sentence "This is the result of a study of more than half a million participants..."
- in paragraph 2, maybe "than" should be used instead of "that" in the sentence "It has been estimated that more than 20_30% of dietary salt..."
Author Response
Reviewer 3: The manuscript deals with the solutions available to reduce salt consumption in various food products. The manuscript is well structured and clearly present the different solutions. A high number of publications is cited.
Many thanks for the positive summary of the review.
I only have few comments : - at the beginning of paragraph 3, maybe add a sentence to remind to the reader what are nudges
Sentence added as suggested.
- in this actual form, the table is difficult to read, maybe it should be better with a higher space between the different publications
I am hoping this formatting issue will be addressed when it comes to production, nevertheless spacing increased as requested.
- in the conclusion, maybe a comment about the use of these different solutions by consumers or industrials may be interesting as some are easy to use while others are more complicated
Comment added as suggested.
- in paragraph 2, I think "is" is missing in the sentence "This is the result of a study of more than half a million participants..."
Added
- in paragraph 2, maybe "than" should be used instead of "that" in the sentence "It has been estimated that more than 20_30% of dietary salt..."
Spelling modified.